# VIM: Variational Independent Modules for Video Prediction

**Rim Assouel**[*]                                                              ASSOUELR@MILA.QUEBEC
**Lluis Castrejon**[*]                                                          CASTREJL@MILA.QUEBEC
**Nicolas Ballas**                                                               BALLASN@FB.COM
**Aaron Courville**                                                   AARON.COURVILLE@UMONTREAL.CA
**Yoshua Bengio**                                                        YOSHUA.BENGIO@MILA.QUEBEC
*Mila, 6666 Rue St-Urbain, Montréal QC*

**Editors:** Bernhard Schölkopf, Caroline Uhler and Kun Zhang

## Abstract

We introduce a variational inference model called Variational Independent Modules (VIM) for sequential data that learns and infers latent representations as a set of objects and discovers modular causal mechanisms over these objects. These mechanisms - which we call modules - are independently parametrized, define the stochastic transitions of entities and are shared across entities. At each time step our model infers from a low-level input sequence a high-level sequence of categorical latent variables that select which transition modules are applied to which high-level object. We evaluate this model in video prediction tasks where the goal is to predict multi-modal future events given previous observations. We demonstrate empirically that VIM can model 2D visual sequences in an interpretable way and is able to identify the underlying dynamically instantiated mechanisms of the generation process. We additionally show that the learnt modules can be composed at test time to generalize to out-of-distribution observations.

**Keywords:** Objects, Modularity, Interpretability, Compositionality, Representation Learning

## 1. Introduction

Predicting future events is believed to be a fundamental function of the human brain (Clark, 2013; Mullally and Maguire, 2014) having implications for representation learning, planning or counterfactual reasoning. Humans have the ability to decompose a visual scene into abstract objects and predict their changes far into the future (Kahneman et al., 1992; Spelke et al., 1993). Most interestingly, humans can adapt to a new situation quickly by re-using relevant prior knowledge about objects and their dynamics. In particular, they are good at generalizing in a compositional way because they represent knowledge as re-usable components that can be further composed to explain new observations.

Recent work (Eslami et al., 2016; Burgess et al., 2019; Greff et al., 2019; Crawford and Pineau, 2019; Locatello et al., 2020) has made significant progress in learning unsupervised entity-centric representations of images and a few of them (Lin et al., 2020a; Kossen et al., 2019; Jiang et al., 2019; Crawford and Pineau, 2020; Kosiorek et al., 2018) have extended those models to generate both deterministic and stochastic videos. However, except for SCOFF (Goyal et al., 2020), to the best of our knowledge none of them have considered architectures that disentangle the underlying dynamic rules of the generation process and rather model the transition part of their models with a shared monolithic module that is applied at each time step. We extend the architectural inductive biases from SCOFF and incorporate them in a probabilistic generative model of videos.

---

[*] Equal contribution.

In this work we propose a variational framework to learn both entity-centric representations and entity-centric transition modules. We show that in a simple 2D stochastic environment we are able to identify the exact generating factors of variation both in terms of declarative (e.g. constitutive objects of the visual scene) and procedural knowledge (e.g. transitions rules that govern the stochastic dynamics of the objects). We argue that having those two layers of structure in a generative model is an essential first step towards generalizing out-of distribution, particularly when the new unseen data results from a composition of seen dynamic rules.

Our model called Variational Independent Modules (VIM), is a probabilistic generative model where both latent states and transitions functions over these latents have an entity-centric inductive bias. It thus learns a latent state composed of a set of abstract entities, or slots (Locatello et al., 2020) and a set of stochastic transition functions over the entities. These transitions functions, which we call modules, are independently parametrized and are shared across entities, following the principle of reusable independent causal mechanisms (Peters et al., 2017; Goyal et al., 2019, 2020). At each time step $t$, our model infers a set of categorical latent variables (called *selection variables* $\mathbf{r}^t$) to select which transition modules to apply to each represented entity in the latent set $\mathbf{z}^t$. We train VIM using variational inference over both the entity-centric set of $K$ latents $\{\mathbf{z}_k\}_{k=1,,K}$ and their respective selection variables $\{\mathbf{r}_k\}_{k=1,,K}$. In particular, the distribution of the categorical selection variables is implemented with a key-query attention mechanism (Bahdanau et al., 2014; Vaswani et al., 2017) in which all the possible modules compete (Goyal et al., 2019; Goyal et al., 2021; Locatello et al., 2020) to explain future states and they are sampled according to their attention importance weights.

One of the key assumptions behind our framework is that the abstract entities are evolving mostly independently and only interact sparsely with each other (Pearl, 2009; Goyal et al., 2019). Consequently, the causal graph is sparse (Bengio, 2017; Goyal and Bengio, 2020) and a module can only consider a handful of slots as input argument. However in this work we consider unary modules and leave the exploration of n-ary modules to model interactions between multiple entities for future work.

We evaluate VIM in a 2D stochastic setting where the goal is to predict multi-modal future events given previous observations. Our contributions are the following:

- We propose a simple stochastic environment in which we know the rules of the dynamics, corresponding to the different modes in the transition distribution, to enable checking for correctly identified solutions (Locatello et al., 2018).

- We show via simulations that our framework is able to identify the ground truth dynamics rules that govern the data generation process in this environment.

- We show that VIM is able to generalize in a compositional and interpretable way in a simple out-of-distribution (OOD) tracking task.

## 2. Background: Recurrent State Space Models

Given a set of $D$ observed frames $\mathbf{c} = (\mathbf{c}_1, ..., \mathbf{c}_D)$ and $T$ following future frames $\mathbf{x} = (\mathbf{x}_1, ..., \mathbf{x}_T)$, our goal is to learn a generative model that maximizes the probability $p(\mathbf{x}|\mathbf{c})$. To solve this task, our proposed model builds upon Recurrent State-Space Models (RSSMs) (Hafner et al., 2019), which we review in this section.

RSSMs define a variational framework to model sequential data. They introduce a sequence of latent variables $\mathbf{z} = (\mathbf{z}_1, ..., \mathbf{z}_T)$ to capture the stochasticity of the observation at each time step such that the joint distribution is factorized as:

$$p(\mathbf{x}, \mathbf{z}|\mathbf{c}) = \prod_{t=1}^{T} p(\mathbf{x}_t|\mathbf{z}_{\leq \mathbf{t}})p(\mathbf{z}_t|\mathbf{z}_{< \mathbf{t}})p(\mathbf{z}_0|\mathbf{c}). \tag{1}$$

where $p(\mathbf{x}_t|\mathbf{z}_{\leq \mathbf{t}})$ is the likelihood model, $p(\mathbf{z}_t|\mathbf{z}_{< \mathbf{t}})$ is the prior transition model and $p(\mathbf{z}_0|\mathbf{c})$ is the discovery model that maps the observed context to a distribution over the initial latent state. The main advantage of RSSMs over previous autoregressive models is computational: it can make multi-step future predictions without having to render/encode observed frames at each time step.

RSSMs are trained using variational inference (Jordan et al., 1999) by introducing an amortized approximate posterior $q(\mathbf{z}|\mathbf{x}, \mathbf{c}) = \prod_{t=1}^{T} q(\mathbf{z}_t|\mathbf{z}_{< \mathbf{t}}, \mathbf{x}_\mathbf{t})p(\mathbf{z}_0|\mathbf{c})$ where $q(\mathbf{z}_t|\mathbf{z}_{< \mathbf{t}}, \mathbf{x}_\mathbf{t})$ is called the posterior transition model and $q(\mathbf{z}|\mathbf{x}, \mathbf{c})$ approximates the true posterior distribution $p(\mathbf{z}|\mathbf{x}, \mathbf{c})$. Training is done end-to-end by maximizing the Evidence Lower Bound (ELBO) (Kingma and Welling, 2013; Rezende et al., 2014).

## 3. VIM: Variational Independent Modules

Similar to (Lin et al., 2020a; Kossen et al., 2019; Jiang et al., 2019; Crawford and Pineau, 2020; Kosiorek et al., 2018), VIM is a probabilistic RSSM where the latent space is structured as a set of $K$ hidden vectors $\mathbf{h} = \{\mathbf{h}_k\}_{k=1..K}$ (e.g. slots (Locatello et al., 2020)) where each slot is supposed to represent an *abstract* entity of the input (e.g. visual objects). In VIM we add an additional layer of structure to account for the fact that only a few rules (Goyal et al., 2020) govern the dynamics of objects in the world and that these rules (that we call *modules*) are shared amongst entities. Our transition distribution $p_\theta(\mathbf{h}_t|\mathbf{h}_{<t})$ is thus parametrized with $M$ independent modules that compete against each other to explain future observations. The modules operate over entities and can have a predefined number of arguments. For our experiments, we only use unary modules that operate independently on a single entity. The high-level computation steps of our model's transition function are the following:

- For each entity $k$ we first compute all the $M$ possible next states given by applying each transition module to its current latent state $\mathbf{h}_t^k$.

- We then use a learned key-query attention mechanism to define a categorical distribution over the possible transitions, represented by the entity-centric module selection variable $\mathbf{r}_t^k$. This attention mechanism compares possible future states to observed states (posterior). We sample the module index according to the distribution over $\mathbf{r}_t^k$.

- Finally we sample a gaussian additive update and update entity states $\mathbf{h}_{t+1}^k$ with the module selection and the additive update. The role of the additive update is to model any remaining stochasticity not captured by the module update.

VIM can thus be interpreted as a RSSM with 2 latent variables, $\mathbf{r} = \{\mathbf{r}_t\}_{t=1..T}$ and $\mathbf{z} = \{\mathbf{z}_t\}_{t=1..T}$ where $\mathbf{r}$ is a module selection variable and $\mathbf{z}$ is a gaussian additive update to hidden

slot representation $\mathbf{h} = \{\mathbf{h}_t\}$. Its generative model is factorized as

$$p_\theta(\mathbf{x}, \mathbf{r}, \mathbf{z}|\mathbf{c}) = \underbrace{p_\theta(\mathbf{z}_0|\mathbf{c})}_{\text{discovery model}} \prod_{t=1}^{T} \underbrace{p_\theta(\mathbf{x}_t|\mathbf{h}_t)}_{\text{observation model}} \prod_{k=1}^{K} \underbrace{\underbrace{g_\theta(\mathbf{h}_t^k|\mathbf{h}_{t-1}^k, \mathbf{z}_t^k, \mathbf{r}_t^k)}_{\text{recurrent update}} \underbrace{p_\theta(\mathbf{z}_t^k|\mathbf{h}_{t-1}^k, \mathbf{r}_t^k)}_{\text{latent update}} \underbrace{p_\theta(\mathbf{r}_t^k|\mathbf{h}_{t-1}^k)}_{\text{module selection}}}_{\text{transition model}} \quad (2)$$

where the observation model $p_\theta(\mathbf{x}_t|\mathbf{h}_t)$ and the discovery model $p_\theta(\mathbf{z}_0|\mathbf{c})$ are both parametrized with a slot-attention network (Locatello et al., 2020). In the following we describe in more details the parametrization for both the inference and generation components of our model.

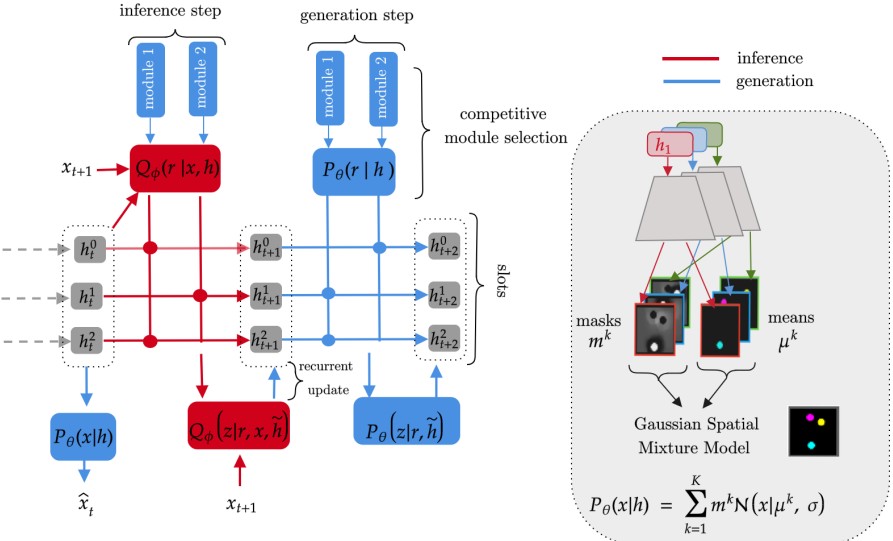

Figure 1: **Overview of VIM** VIM disentangles both Objects and Dynamics with slots and transition modules. *Left*: structure of the transitions, controlled by the selection of an inferred rule $r$ and its application to the latent state $z$, and by an observation model of $x$. *Right*: generative decoder architecture mapping a set of object descriptions $z$ to an image where the different objects are rendered and composed using a Spatial Guassian Mixture Model.

### 3.1. Generation

We denote the representation of slot $k$ at time-step $t$ as $h_t^k$. We initialize slots as $h_0 = \mathbf{z}_0$ and update them recurrently at each time-step: $h_t^k = g_\theta(h_{t-1}^k, \mathbf{z}_t^k, \mathbf{r}_t^k)$. We also introduce the set of $M$ possible future candidates for each entity $k$ and denote it $\mathbf{s}_t^k = \{f_{\theta_m}(h_{t-1}^k) = f_\theta(h_{t-1}^k, \mathbf{r}_t^k = m)\}_{m=1..M}$ where the modules are parametrized with $M$ independent MLPs whose parameters are indexed by $\{\theta_{m=1..M}\}$.

**Module Selection Prior** In this step we describe the parametrization of the module selection prior $p(\mathbf{r}_{t+1}^k|h_{<t})$ where $\mathbf{r}_{t+1}^k$ defines a categorical variable that indexes which module to apply to entity $k$ at time step $t$. The probability of the categorical distribution from which $\mathbf{r}_{t+1}^k$ is obtained with

a key-query attention mechanism where the keys are the $M$ candidates $\mathbf{s}_t^k$ and the query are the current hidden states $h_t^k$. We sample the module $m$ and update the candidate hidden states using a Gumbel softmax (Jang et al., 2016).

**Latent Update Prior**    The latent update is the result of two additive updates followed by a layer normalization such that at each time step $t$ we recurrently update slot $k$: $\mathbf{h}_t^k = g_\theta(\mathbf{h}_t^k|\mathbf{h}_{t-1}^k, \mathbf{z}_t^k, \mathbf{r}_t^k) =$ LayerNorm$(\mathbf{h}_{t-1}^k + \mathbf{z}_t^k + f_\theta(h_{t-1}^k, \mathbf{r}_t^k))$ where $\mathbf{z}_t^k$ is sampled from a gaussian prior whose mean and variance are computed as follows: $\mu_k^t, \sigma_t^k = \text{MLP}(f_\theta(h_{t-1}^k, \mathbf{r}_t^k))$ and $\mathbf{z_t^k} \sim \mathcal{N}(\mu_k^t, \sigma_t^k)$

### 3.2. Inference

In this section we describe the parametrization of our inference model, which is factorized as:

$$q(\mathbf{z}, \mathbf{r}|\mathbf{x}) = \prod_{t=1}^{T} \prod_{k=1}^{K} q_\phi(\mathbf{z}_t^k|f_\theta(\mathbf{z}_{<t}^k, \mathbf{r}_t^k), \mathbf{x}_t) q_\phi(\mathbf{r}_t^k|\mathbf{z}_{<t}, \mathbf{x}_t). \tag{3}$$

During inference we need to encode information about the target frame to infer the updates to apply to each entity slot $k$. However not all the visual information is needed for all the abstracted entities. Each entity needs to attend to a specific perceptual grouping of the target frame. To that end, we propose to use a slot-attention mechanism where the grouping is learned through a key-query attention mechanism. We denote $\mathbf{x}_t^k$ the perceptual grouping for entity $k$ obtained at time step $t$ such that :

$$\hat{\mathbf{x}}_k^t = \sum_i \beta_{k,i}^t V_i^t \text{ with } \beta^t = \text{softmax}(\frac{KQ^T}{\sqrt{D}}, \text{dim="slots"}) \tag{4}$$

where the keys $K$ are extracted from the current states $\{h_t^k\}_{k=1..K}$ and the queries and values are extracted from the target input $\mathbf{x}_t$, encoded with a size-preserving CNN backbone and a positional encoding.

**Module Selection Posterior**    We define a rule selection posterior $q_\phi(\mathbf{r}_t^k|h_{<t}, \mathbf{x}_t)$ for each entity $k$. Like with the module selection prior in generation, the idea is to score the set of candidates with a key-query attention mechanism where this time the query for entity $k$ is obtained from the entity-centric target encoding $\mathbf{x}_t^k$. The intuition behind this design choice is to select the update that best explains the current observation. We sample the module $m$ to apply with a Gumbel softmax trick.

**Latent Update Posterior**    The latent update posterior is similar to the prior, where this time the gaussian parameters of $\mathbf{z_t^k}$ are computed as a function of an entity-centric target encoding $\mathbf{x}_t^k$ such that: $\hat{\mu}_k^t, \hat{\sigma}_t^k = \text{MLP}([f_\theta(h_{t-1}^k, \mathbf{r}_t^k); \mathbf{x}_t^k])$ and $\mathbf{z_t^k} \sim \mathcal{N}(\hat{\mu}_k^t, \hat{\sigma}_t^k)$.

### 3.3. Training

Training is done using variational inference maximizing the following evidence lower bound:

$$\mathcal{L}(\theta, \phi) = \sum_{t=1}^{T} \mathbb{E}_{q_\phi}[\log p_\theta(\mathbf{x}_t|\mathbf{z}_t, \mathbf{z}_{<t})] - \underbrace{KL(q_\phi(\mathbf{z}_t|\mathbf{r}_t, \mathbf{x}_t, \mathbf{z}_{<t})||p_\theta(\mathbf{z}_t|\mathbf{r}_t, \mathbf{z}_{<t}))}_{\text{gaussian KL}}$$

$$- \underbrace{KL(q_\phi(\mathbf{r}_t|\mathbf{x}_t, \mathbf{z}_{<t})||p_\theta(\mathbf{r}_t|\mathbf{z}_{<t}))}_{\text{module KL}}$$

where there is a tension between explaining the multi-modality of the future outcomes either using the gaussian distribution of a single module (but this would mean trying to fit a unimodal gaussian distribution to multimodal observations and would result in a high noise KL) or using several modules indexed by $\mathbf{r}$ to explain the different modes. The module KL also prevents the model to learn duplicate transition modules and we verify this experimentally by training the model with more modules than actual modes in the distribution. In that case, additional modules are simply rarely selected during inference.

## 4. Related Work

**Video Prediction**   Since early work inspired by language modelling (Ranzato et al., 2014; Srivastava et al., 2015), video prediction has seen great progress leveraging advances in generative models and deep learing architectures. The temporal and spatial dependencies between pixels is typically captured via autoregressive models (Larochelle and Murray, 2011; Dinh et al., 2016; Kalchbrenner et al., 2017; Reed et al., 2017; Weissenborn et al., 2020) or latent variables models such as the VAE (Kingma and Welling, 2013; Chung et al., 2015; Denton and Fergus, 2018; Lee et al., 2018; Castrejon et al., 2019; Villegas et al., 2017a,b) or GAN  (Goodfellow et al., 2014; Vondrick et al., 2016; Mathieu et al., 2015). Our approach uses variational inference for training, like other VAE-like models. Most of those previous models, however, rely on fixed-size unstructured vectorial representations of the state and monolithic prediction models while we explore the use of structured latent space (as a set of slots) and modular architectures (for the mechanisms) to explain multimodality in the transitions.

**Unsupervised Object Discovery**   A recent research direction explores unsupervised object-centric representation learning from visual inputs. The main motivation behind this line of work is to disentangle a latent representation in terms of objects composing the visual scene. They can be divided into two types of models: on one hand, the spatial mixture models (Locatello et al., 2020; Burgess et al., 2019; Greff et al., 2019) learn a set of unstructured latents that are decoded into a pixel-wise mean and mask such that each pixel location defines a Gaussian mixture model weighted by the slot masks. On the other hand, spatial-transformer based models propose to further disentangle each slot latent representation into several variables (e.g. content, location, presence, depth) where the location variable parametrizes the input to a spatial transformer that fills a canvas, as in (Eslami et al., 2016; Crawford and Pineau, 2019; Stelzner et al., 2019; Lin et al., 2020a). The contribution of our work is orthogonal and we want to show the benefits of disentangling the mechanisms that process the slots in the transition module of a sequential generative model when the target distribution has multimodal uncertainty. Lin et al. (2020a) argues that a careful design of the prior is needed to account for multimodal future trajectories and propose to do so with a slot-wise hierarchical Gaussian prior model. In this work we propose an *interpretable* alternative to account for several modes

in the target distribution and show that a modular approach is capable of capturing the underlying factors (e.g. modes in our case) of the generating process. In terms of architectural components, we extend Locatello et al. (2020)'s slot attention module to a sequential setting where the slot attention module is used to encode a meaningful part of the target frame in a slot-wise manner. We show that by using independently parametrized modules in the recurrent transition function, our approach is able to discover object-centric dynamical rules in an unsupervised manner and that the slot attention inference machinery is able to select compositions of those dynamical rules at test time.

**Independent Mechanisms**   Recent approaches have explored architectures composed of a set of independently parametrized modules which compete with each other to communicate and attend or process an input (Goyal et al., 2019; Goyal et al., 2021). Those architectures are inspired by the notion of independent mechanisms (Pearl, 2009; Bengio et al., 2019; Goyal et al., 2019), which suggests that a set of independently parametrized modules capturing causal mechanisms should remain robust in case of distribution shift due to an intervention, as adapting one should not require adapting the other modules. The recurrent independent mechanisms architectures (Goyal et al., 2019; Goyal et al., 2021) however are not probabilistic models; they cannot capture the uncertainty inherent with future predictions. In this work we formulate the same intuition of mechanisms separation in a variational inference framework where the selection of the the mechanisms is expressed as a categorical random variable whose posterior distribution the model must infer. We further showcase an interpretability advantage in the case of a simple 2D stochastic environment.

## 5. Experiments

In this section we describe experiments designed to showcase three main properties of VIM: i) object-centric *multi-modality* of the predictions in stochastic environments, ii) *interpretability* of the learned modules, and iii) *generalization* to compositional out-of-distribution settings in an interpretable way.

### 5.1. Experiment 1 - Interpretability

In this section we show that the modular architecture of VIM is able to capture the underlying generating factors of the target distribution in an interpretable way. Through an ablation study on the number of available modules we also show that with enough modules to capture the different distribution modes precisely, VIM performs better than models that rely on a unimodal prior.

**Dataset**   We evaluate VIM on a dataset built to showcase interpretability of the modules. This dataset, called the **Random Walk Dataset** consists of 2D image sequences of multiple colored-balls that evolve on a black background. At each time step each ball can move randomly in one of the 4 cardinal directions. The resolution of the videos is $32 \times 32$ and the model is trained on up to 15 consecutive frames.

**Setup**   We train VIM with $N$ modules and $K = 4$ slots on trajectories that contain 3 balls. During training, the selection variable $\mathbf{r}$ is sampled using a Gumbel-softmax (Jang et al., 2016) with a fixed temperature of 1 and we use the hard version during testing so that the variables are categorical. For evaluation we match each ground truth ball of the visual scene to a latent slot explaining it. Note that Spatial-Transformer based models (Jang et al., 2016; Crawford and Pineau, 2019) directly exploit the slot bounding box center coordinate to compute this matching. Our model does not consider

object bounding boxes. Instead, we use the slot-wise masks outputted by the observation model to match them to objects. More specifically, to determine the location $(x_k, y_k)$ of the object explained by the $k$-th slot we sample coordinates uniformly over the 2D space covering the whole frame and then we average these coordinates weighted by the value of the slot mask value at each location $m_k^{x,y}$, such that:

$$x_k, y_k = \frac{\sum_{(x,y) \in [1,H] \times [1,W]} (x, y) * m_k^{x,y}}{m_k^{x,y}}.$$

Once we have those slot-wise coordinates we compare them against ground truth balls positions and match a slot with the closest ball position in the first frame and keep the same matching for the rest of the sequence. We then use these matched pairings to compute ball position errors for tracking purposes or to compare the slot-wise module selection variable to the ground-truth action transition.

**Results** Our objective is to verify that the learnt modules can be directly interpreted as causal dynamics rules that underlying the generation process for the Random Walk dataset. We show that each module corresponds to a particular mode of the generating distribution (e.g. moving in one of the cardinal directions). To do so, we evaluate our inference model on some test sequences of length 20 and store the selection variables of all the slots $((\mathbf{r}_k^t)_{k=1..K})_{t=1..20}$. Using the first set of slots for the sequence we match each ground truth ball with its explaining slot, whose location has been obtained with its decoded mask by choosing the closest one. We denote the slot associated with the $d$-th ball by $i_d$. We compare at each time step $t$ the ground truth transitions of each ball $d$ with the index of the module that slot $i_d$ has selected and report the proportion of the correct correspondences in Figure 2.

We further visualize the repeated effect of each module on a single ball to confirm the correspondences. Figure 2 shows the result of applying different modules to given slots. We observe that each module corresponds to a particular movement direction, matching the possible ground-truth moves. When the model is trained with more modules than dynamic rules in the environment, then these superfluous modules are not used. This behaviour is in part encouraged by the module KL term of the loss which will be larger if 2 modules are identical, as modules that explain the same mechanisms would have the same selection probability. We show in Figure 2 the correspondence between selected modules and ground truth actions for a VIM model with 5 modules. In this case, only 4 modules appear to be selected by the slots that represent ball entities.

**Effect of the number of Modules** In this section we study the effect of the number of available modules to model the sequences when generating video sequences. The model is trained on the Random Walk dataset with a varying number of modules. Larger per-module capacity is given to models with less modules to factor out the effect of total number of parameters in the modeling ability. In Figure 3 we compare three regimes: a monolithic transition function with a single module, transition functions with less modules than actual modes in the distribution, and a transition functions with more modules than actual modes in the distribution. The monolithic transition model can only rely on the unimodal Gaussian distribution to capture the multimodal stochasticity of the transitions, which results in lesser ability to model sequences. In the low modules regime, the model separates the modules to explain the modes in a hierarchical way, but some modes are shared between modules which results in not sufficient modeling capacity. In the case of 5 modules, module number 1 is never selected and thus doesn't appear in the color map. In this setup where there are enough modules to represent all the modes of the transition function, our model learns interpretable

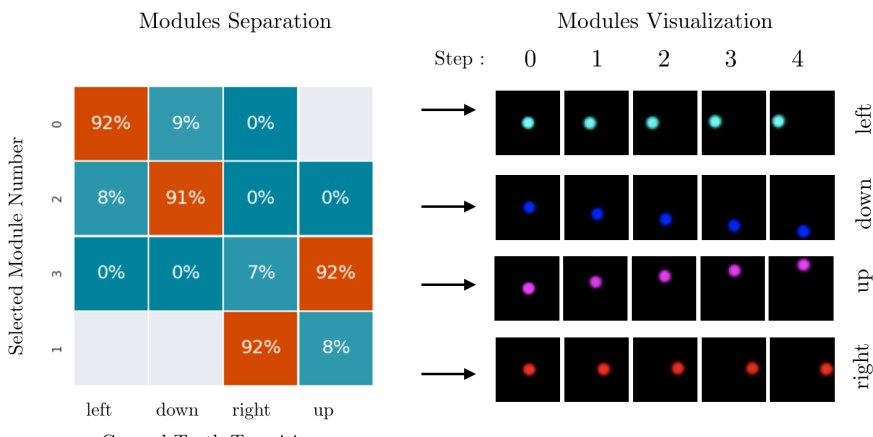

Figure 2: **Dynamics Rules Discovery** The proposed VIM model learns to match modules to ground truth entity transitions. *Left:* For each rule, we show the distribution over selected modules for a 4-module model. Modules specialize to implement one rule, with additional modules not being used. *Right:* We show the effects of repeatedly applying a module to a slot. Module transitions are interpretable and correspond to ground-truth transitions.

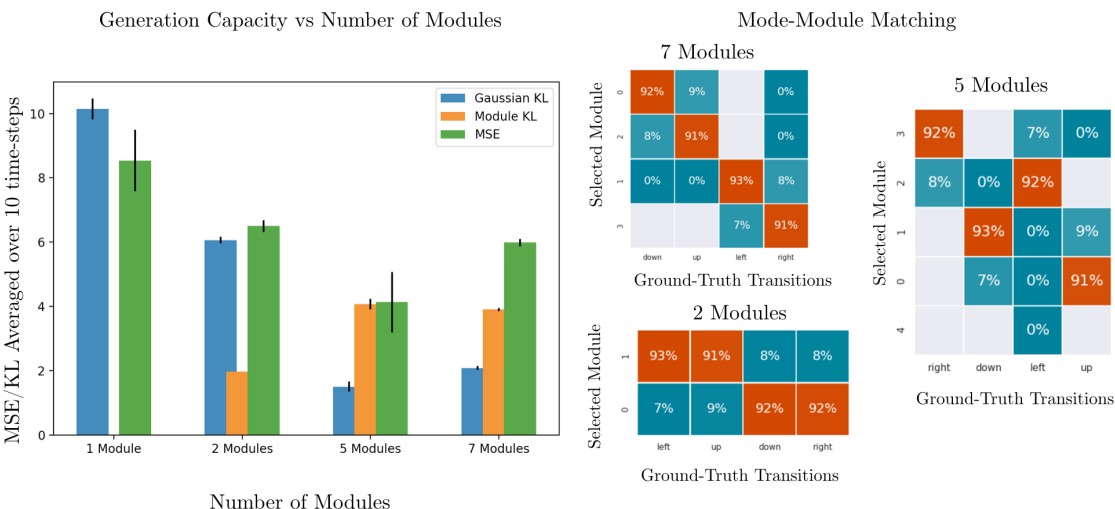

Figure 3: **Modules number ablation** *Left:* MSE/KL of the inference model averaged over 10 time steps for VIM with a varying number of modules *Right:* Modules specialize to each explain a mode in the distribution when enough modules are available.

rules and correctly models the data sequences. When VIM has more modules than actual modes in the distribution, additional (and non useful) modules are almost never selected. We show the correspondence between modules and rules in Figure 3.

## 5.2. Experiment 2 - Compositionality

**Dataset**    In this section we introduce a variant of the Random Walk dataset where the coloured balls dynamics at each step is a composition (e.g. left-up-up-left) of the previous atomic cardinal transitions. Between two consecutive frames, we sample the sequence of modules that composes the transition of each ball at random. When trained on this dataset, the model never sees the the atomic transitions that compose the random walk of each ball. We call it the Compositional Random Walk.

**Setup**    We introduce a variant of VIM where at each time step we allow the model to perform multiple selections and update steps before rendering. The rationale behind this variant is that for sequences whose transition dynamics rules are compositional we want to check whether VIM is able to compose some atomic building blocks (e.g. the modules) to explain an observation both at training and testing time. When the model is allowed $k$ iterations before rendering we simply denote the variant VIM-$k$.

**Results**    The insights we derived from the results of these experiments are two-fold:

- When trained on the atomic setting, VIM generalizes in an interpretable manner to compositions of transitions seen during training.

- When trained on the compositional setting, the individual modules that are learned in the multi-step version of VIM are still interpretable in terms of atomic transitions that were never seen during training.

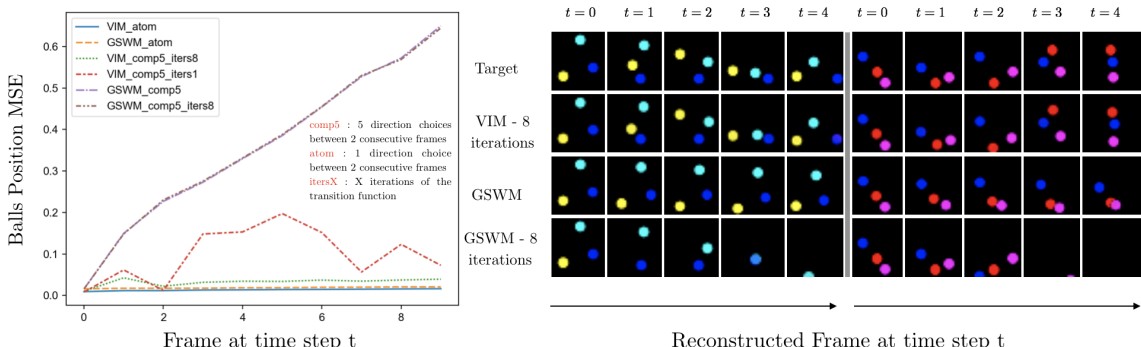

Figure 4: **OOD Tracking**. VIM's ability to track to out-of-distribution composition of transitions seen during training. Compositional dynamics tracking comparison of VIM and G-SWM and their respective multi-step versions. Both trained on the atomic setting and are compared qualitatively *Right*: and quantitatively *Left* reporting their respective ball position errors.

**OOD Compositional Generalization**    We train VIM on the atomic setting and test it on out-of-distribution compositional sequences where up to 5 choices of atomic transitions are allowed for each ball between two consecutive frames.(e.g. *left-down-right-down-right*). At test time we consider a multi-step version of VIM where 8 iterations of the selection-update step are allowed in between frames. We show that the inference key-query selection part of VIM acts at each iteration as a kind of planner in the space of modules and selects the module that best explains the current observation. In Figure 5 we render the resulting frame at each iteration and notice that VIM composes

modules in an iterative manner such that each ball progressively gets closer its target position before oscillating around it. Additionally, in Figure 4 we compare the tracking ability of a state-of-the art object-centric video generation model, G-SWM (Lin et al., 2020b), with this multi-step version of VIM . We show that both the key-query selection mechanism and the composable modules enables VIM to better track objects that have a different amplitude than the one seen during training. For a fair comparison we also consider a multi-step version of G-SWM where several iterations of the same monolithic propagation module is allowed between 2 consecutive frames.

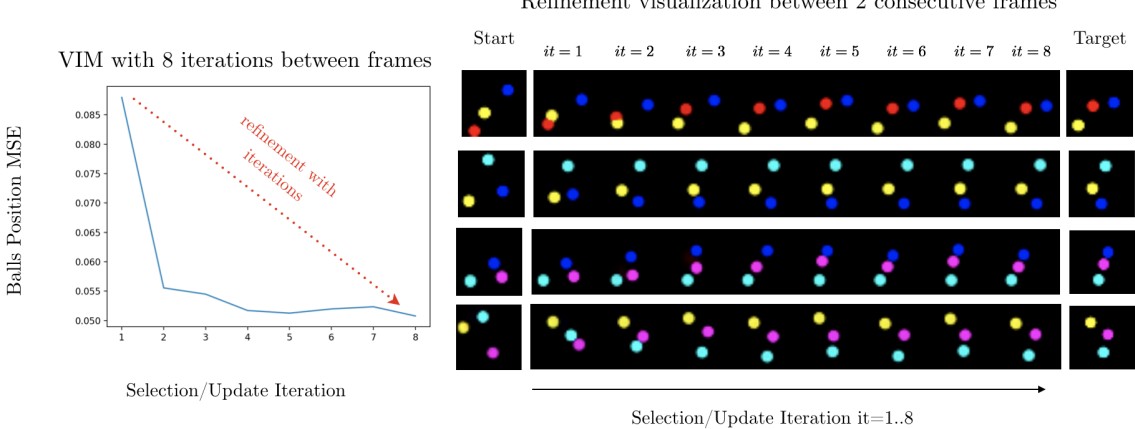

Figure 5: **Iterative Refinement**. We show that VIM can model out-of-distribution compositions of transitions. *Left:* VIM trained on the Random Walk Dataset and tested on the Compositional Dataset allowing 8 iterations between frames. Between 2 consecutive frames, the ball position error decreases with the number of iterations. *Right*: Visualization of the module iterations. VIM iteratively selects modules to modify the starting observation and reconstruct the target frame.

**Multi-step Interpretability** We train a 2-step version of VIM on the compositional setting and restrict the number of modules to 5 so that we still have more modules than atomic transitions (that are never seen) but less modules than possible compositional transitions seen during training (e.g. *left-right, up-left, down-down*, and so on). We test this model on sequences of atomic transitions to evaluate the proportion of selected modules corresponding to each ground truth transition. We show in Figure 6 that the same modularisation emerges as when VIM is trained on the atomic setting. In this compositional setting, when evaluated on atomic transitions we observe the same module specialization as when it was trained on atomic transitions. This shows that when trained with 2 selection/update iterations between 2 consecutive frames VIM successfully identifies the underlying atomic transformations that compose the transitions VIM has been trained on without directly observing them.

### 5.3. Experiment 3 - Dynamics Transfer

In this section we are interested in showing that VIM learns a factorized representation formed by entities and transition rules over these entities. In particular we would like to evaluate whether VIM is able to recognize known dynamics applied to unseen objects and that the inference machin-

Figure 6: **Rule-Module matching for compositional transitions** We show a rule-module matching histogram for a VIM model with 5 modules and 2 module iterations per frame trained on the Compositional Random Walk dataset. We observe that, despite only observing compositions of actions, the modules implement and match ground-truth atomic transitions.

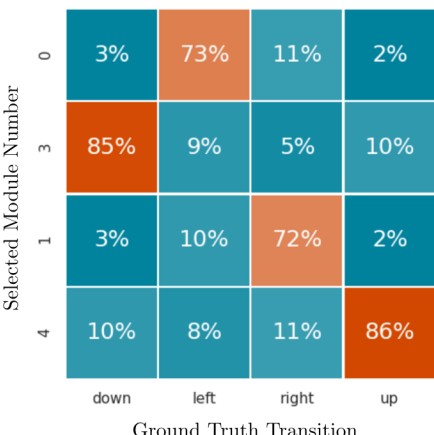

ery proposes to transfer those dynamics to known objects using the selection variables for these dynamics.

**Dataset** In this section we test VIM on a dataset where objects have out-of-distribution attributes (e.g. shape, color, size), beyond those seen during training, but following the same Random Walk dynamics seen during training: they can go in one of the four cardinal directions between two consecutive frames and with the same amplitude.

**Results** We train VIM on the Random Walk Dataset with coloured balls of the same size and test it on a OOD version where objects have different shape, colour and size than the balls seen during training. During testing, we run the inference model on these OOD target sequences $(\mathbf{x}_t^{target})_{t=0..T}$ and extract the selection variables at each time step $(\mathbf{r}_t^{target})_{t=1..T}$. We then consider an image $c$ where balls with known attributes are placed at the exact same location as the first frame of the target sequence and transfer the extracted target dynamics such that the rendered sequence is sampled following :

$$\{\mathbf{x}_t^{transfer}\}_{0:T} \sim \prod_{t=1}^{T} p(\mathbf{x}_t^{transfer}|\mathbf{z}_t) \prod_{k=1}^{K} p(\mathbf{z}_t^k|f(\mathbf{z}_{t-1}^k, \mathbf{r}_t^{target,k}))p(\mathbf{z}_0^k|\mathbf{c}).$$

In Figure 7 we show that VIM is able to recognize known dynamics of objects that have OOD attributes. VIM has successfully factorized individual objects identity and the shared dynamic rules that are applied in the same way irrespective of the objects identity.

### 5.4. Experiment 4 - Comparison with SCOFF/RIM

In this section we perform a comparison to SCOFF (Goyal et al., 2020) and RIM (Goyal et al., 2019). A key assumption in these models is that the outcome is deterministic given previous frames. We argue that this leads to blurry predictions in settings with stochastic dynamics, as the model is trying to capture the mean of all plausible future outcomes. VIM incorporates the modelling assumptions of SCOFF in a variational inference model that can capture stochasticity in future observations.

We compare to SCOFF and RIM when modeling the Random Walk dataset of Experiment 1. We report in Table 1 the MSE averaged over 5 time-steps sequences for SCOFF and RIM (with teacher forcing at each time step) and VIM (using its inference network), both trained with 5 modules and 4

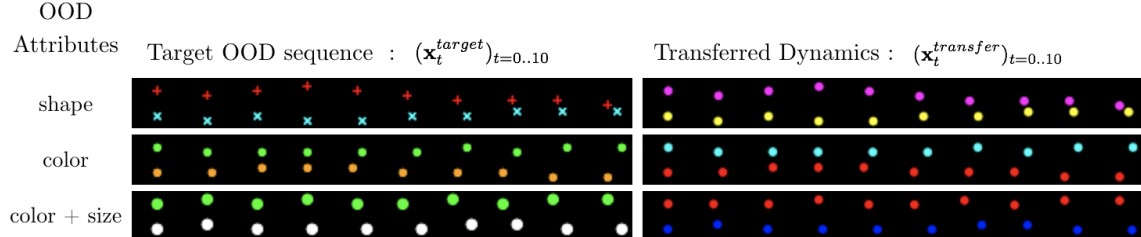

Figure 7: **Dynamics Transfer with OOD Objects**. VIM is able to recognize and transfer known dynamics of novel objects with OOD attributes. *Left*: Target trajectory from OOD objects from which we extract the module selection variables. *Right*: Transferred dynamics on known objects by applying the same selected modules as for the target sequences.

| Model | Reconstruction MSE ($\downarrow$) |
|---|---|
| RIM (Goyal et al., 2019) | $91.7 \pm 5.4$ |
| SCOFF (Goyal et al., 2020) | $81.9 \pm 1.0$ |
| VIM (Ours) | $\mathbf{4.2 \pm 0.5}$ |

Table 1: **Comparison to RIM and SCOFF:** We compare our model to RIM and SCOFF. All models use 4 slots/object files, and SCOFF and VIM use 5 modules/schemata. We report the reconstruction MSE for sequences of 5 frames. The ground-truth previous frame is given to SCOFF/RIM at each time step. Our model produces significantly better reconstructions of the input sequences. Since these sequences are stochastic, RIM and SCOFF do not properly model them as they use deterministic dynamics.

slots. We expect SCOFF to fail to produce sharp samples since it assumes the future is deterministic given the previous frame. whereas we expect VIM to be able to handle the multimodality of the future outputs. For further details about this experiment as well as a visualization of the result please refer to the Appendix.

## 6. Conclusion

VIM introduces a variational framework to disentangle both the constituent entities (slots) and the generating factors (rules) of object dynamics in stochastic scenes. We show that the modular architecture of VIM captures different dynamics rules in an interpretable manner. Moreover, VIM modules can be composed at test time to explain OOD dynamics not seen during training. Additionally, we show that VIM is able to recognize the dynamics of objects with OOD attributes and transfer them to known objects by re-using the learned modules on different slots. The work presented here could be extended in different ways. For example, we could add interactions between slots through the use of n-ary modules. We could additionally scale up the model architecture and capacity to more complex datasets with in-the-wild 3D scenes and test its performance on downstream tasks. We plan to explore these directions in follow-up work.

## Acknowledgments

We thank Facebook AI Research and Mila Quebec AI Institute for managing the computer clusters on which this research was conducted. This work was supported by an IVADO PhD Fellowship to L.C., an Antidote PhD Fellowship to R.A and by funding from CIFAR and NSERC.

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

## Appendix A. Implementation and Experimental Details

| Type | Size/Channels | Activation | Other |
|---|---|---|---|
| Conv $5 \times 5$ | 64 | ReLU | stride 1 |
| Conv $5 \times 5$ | 64 | ReLU | stride 1 |
| Conv $5 \times 5$ | 64 | ReLU | stride 1 |
| Conv $5 \times 5$ | 64 | ReLU | stride 1 |
| Positional Encoding | 64 | - | LayerNorm(64) |
| Linear | 64 | ReLU | - |
| Linear | 64 | ReLU | - |

Table 2: Slot Attention CNN Encoder

| Type | Size/Channels | Activation | Other |
|---|---|---|---|
| Spatial Broadcast | $H \times W$ | - | - |
| Positional Encoding | 64 | - | - |
| Conv $5 \times 5$ | 64 | ReLU | stride 1 |
| Conv $5 \times 5$ | 64 | ReLU | stride 1 |
| Conv $5 \times 5$ | 64 | ReLU | stride 1 |
| Conv $5 \times 5$ | 4 | ReLU | stride 1 |
| Split Channels | RGB(3) masks (1) | softmax on masks | - |
| Recombine slots | - | - | spatial mixture |

Table 3: Broadcast Decoder

| Type | Size/Channels | Activation | Other |
|---|---|---|---|
| Linear | 128 | ReLU | - |
| Linear | 128 | - | Layernorm(128) |
| Candidates | - | - | 1 module example |
| Linear | 128 | ReLU | - |
| Linear | 64 | ReLU | - |
| Stochastic update : | - | - | - |
| Linear | 64 | ReLU | - |
| Linear | 128 | - | - |
| Chunk | Mean (64) Std (64) | - | - |

Table 4: Transition Modules

### A.1. Key-query Selection

The key and query network that are used for the module selection bottleneck are implemented with simple 2 layer MLP of hidden sizes [64, 64] and a ReLU activation on the first layer. We used a fixed temperature of 1 for the Gumbel-Softmax computation.

### A.2. Training Schedule

In our model, the slot attention module is first pre-trained on the first images of the videos following the same learning rate warmup suggested in (Locatello et al., 2020) for about 100 epochs to obtain initial slot separation. We found that it stabilized and speeded up the training on the rest of the videos sequences. We trained the model on sequences of length 20 using a length schedule that starts at 6 frames and increase the number of frames by two every 40 epochs.

## Appendix B. Comparison with SCOFF/RIM - Additional details

SCOFF (Goyal et al., 2020) proposes an architectural component to process a visual input such that it can be explained in terms $K$ object files (corresponding to slots in our model) and $M$ schemata (modules) processing those objects files. The proposed model is tested several downstream tasks including video prediction. To do so they train SCOFF on a next-step prediction task with an autoregressive formulation. RIM (Goyal et al., 2019) corresponds to a SCOFF with $M = 1$ schema (in other words, with a single module). At each time step $t$ the computations are conditioned on the previous frame. The model is trained using teacher forcing, and its training setup can be summarized as follows:

1. Each object file $h_{t-1}^k$ competes to attend to a part of the previous input frame (similar to slot attention) $x_{t-1}^k$.

2. For each object file, the $M$ possible slot candidates given by the application of each schemata are computed.

3. Each object file selects the most plausible candidate with a key-query attention mechanism and updates its state.

4. Each updated object file is then decoded to reconstruct the next-frame.

We show in Figure 8 a reconstructed sequence with a SCOFF model trained with 4 slots and 5 schemata on our Random Walk Dataset. We notice that:

• The object files capture the different objects in the scene, as slots do for VIM.

• Each object file attends to the part of the input that belongs to its represented object.

• The reconstructed frames show the mean position in all 4 possible directions, instead of capturing different directions with different schemata.

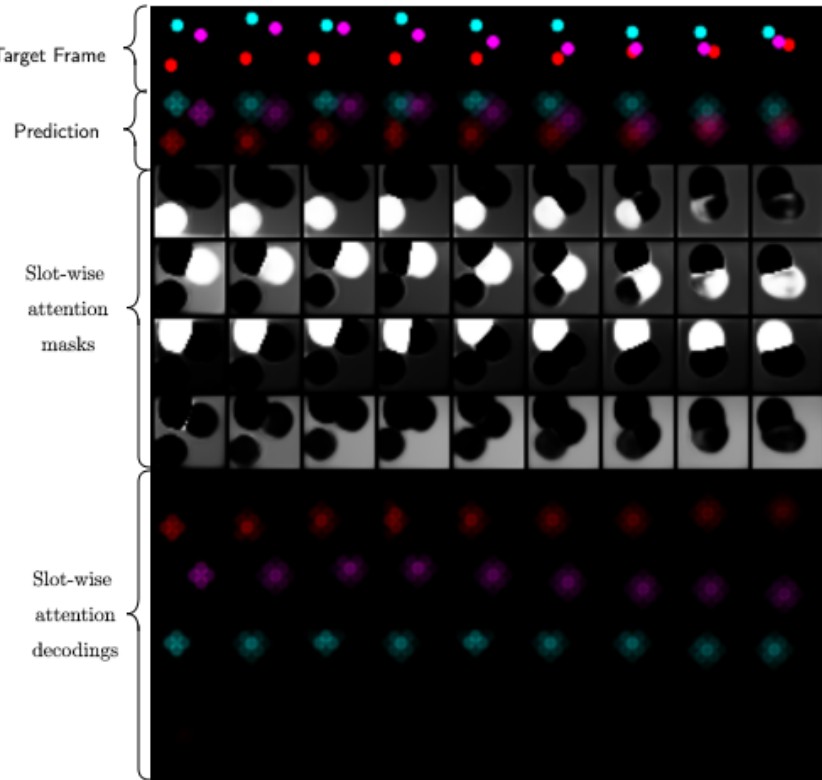

Figure 8: **Sample from SCOFF:** We show a sample of SCOFF in our proposed dataset. This sample illustrates the performance of RIM and SCOFF on this dataset. We observe that the slots correctly segment the input objects and model them as distinct entities. However, the model fails to produce sharp next step predictions, and instead produces the mean of all possible future outcomes from the previous frame. This is because RIM and SCOFF are deterministic models. Note that here we are showing sequences using teacher forcing.

