# OpenReview forum: "VIM: Variational Independent Modules for Video Prediction"
_cclear.cc/CLeaR/2022/Conference — CLeaR 2022 Poster_

### Official Review · Reviewer_hQZF · 2021-11-08

**Confidence:** 3
**Overall Score:** 6

**Main Review:**

The research question that this paper tries to solve is highly relevant for the research field of object-centric representation learning and bringing it closer to the causality domain. The initial idea of the paper is to combine an existing architecture, SCOFF (Goyal et al., 2020), with object-centric video representation learning. However, the model section contains several unclarities which makes it hard to fully grasp the proposed method:
* The latent update prior is said to model the distribution $p(z^{k}\_{t+1}|f(z^{k}\_{t},r^{k}\_{t+1}))$. However, $z^{k}\_{t+1}$ is defined as the output of a Layer Normalization later, with some noise added to the input of the normalization layer. How does the distribution $p(z^{k}\_{t+1}|f(z^{k}\_{t},r^{k}\_{t+1}))$ actually look like then? Why is the noise sampled instead of modeling an actual distribution? Is the distribution not needed in the Gaussian KL term?
* How is the distribution $q(r|x)$ in Equation 5 defined? In Section 3.2, a module selection posterior is mentioned that is of the form $q(r\_t|z\_{t-1},x)$. Is $q(r|x)$ marginalized out over $z$? Or is it a different network?
* The model selection prior $p(r_{t+1}|z_t)$ seems to be independent of previous module selections $r\_{\<t}$. Why is that? Is the module newly chosen at every time step, and not consistent across the time series?
* Figure 1 shows that $p(x|z)$ is parameterized as a mixture of Gaussians, but images are naturally discrete and/or have a finite set of possible pixel values. Is a mixture of discrete logistics used as replacement, as done in e.g. Salimans et al. (2017)? Or how is a degenerated solution prevented?
* The module selection resembles the mixture of expert setup used in other network architectures, like Fedus et al. (2021). However, the drawback of the proposed method is that all modules need to be run on the all inputs, even during inference, instead of only running the selected modules. Why is that required? Why isn't a learnable key for each module used without the need of running each module during inference?
* In Section 5.1, paragraph Setup, it is mentioned that only during testing, the selection variable $r$ is categorical. Does this mean that during training, $r$ is a one-hot vector with values between 0 and 1, or is it still forced to be a one-hot vector and uses the straight-through gradient estimator?
Considering that no code was submitted, it would be hard to reproduce this architecture.

Another unclarity of this paper is with respect to the motivated inductive bias. In the introduction, the setup of the latent space is stated as the inductive bias. However, the inductive bias actually comes from the fact that the decoder is limited in expressiveness, such that sticking with a single module cannot achieve the same performance. Thus, the proposed model is limited to the setting where given the module, the output distribution is known/simple (e.g. a Gaussian). If more complex distributions would be possible in the output space (e.g. a mixture of logistics), the proposed model would not work anymore. Further, if the input image $x_{t-1}$ would already give clues about the module to use, it can collapse again since $z_{t-1}$ would be sufficient to predict $z_{t}$. Similarly, multiple objects with different appearances and different sets of actions could use the same modules for different actions, degrading the interpretability aspect. Thus, the use cases of this model seem very limited.

The experiments are focused on a single dataset which was created for the purpose of this paper. While it showcases the working of the model, it is very simplistic. The fact that most modules are correctly selected in only 4 out of 5 cases on average (i.e. 80%) is not entirely convincing, considering the simplicity of the dataset. A more realistic dataset should be used to showcase this, although it might be hard to find one considering the drawbacks mentioned above.

In conclusion, while I could see the paper being a valid, small contribution to the object-centric representation learning research field, the current form of the paper contains too many unclarities to fully understand the method, and the use cases seem limited.

#### Minor comments
* Figure 1: the colors in the legend for inference vs generation class with the chosen slot colors in the picture right below. It suggests that $z_1$ belong to inference, and $z_2$ to generation.
* The citations should be checked for using the correct journal/conference (e.g. Locatello et al. 2020 was published in NeurIPS)

#### References
* Fedus, William, Barret Zoph, and Noam Shazeer. "Switch transformers: Scaling to trillion parameter models with simple and efficient sparsity." arXiv preprint arXiv:2101.03961 (2021).
* Salimans, Tim, et al. “PixelCNN++: Improving the PixelCNN with Discretized Logistic Mixture Likelihood and Other Modifications.” arXiv preprint arXiv:1701.05517 (2017).
* Goyal, Anirudh, et al. "Object files and schemata: Factorizing declarative and procedural knowledge in dynamical systems." arXiv preprint arXiv:2006.16225 (2020).

__Post-rebuttal changes__
The rebuttal of the authors clarified several aspects of the paper. Under the assumption that these clarifications will be included in the final version of the paper, I have updated my score.

**Summary:**

This paper proposes an extension of current object-centric representation learning method to group transition dynamics per object in a video into multiple independent modules. The model is trained in a generative fashion where the changes in latent space over time are parameterized by those modules.

---

> ### Author Response · Authors · 2021-12-03
> **Reply**
>
> Thank you for your insightful comments. We realize there were confusing terms in our model explanation. We have updated the model description section to be more consistent and clear (see general comment, and [updated model](https://postimg.cc/ZBJTNHTn) ). We answer your questions in more detail below:
>
> * Latent/additive update distribution
>
> Our model has two random variables: the module selection variable and an additive update to the slot representation. The module selection variable $\mathbf{r}$ is discrete, whereas the additive update  $\mathbf{z}$ to the slot representation  $\mathbf{h}$  is continuous. $\mathbf{z}$ follows a gaussian distribution. The slot representation $\mathbf{h}$ is the result of i) updating the previous slot representation with the selected module $\mathbf{r}$, ii) adding the additive update $\mathbf{z}$ and iii) normalizing the result (see general comment for formulas). Therefore, the distribution of the slot representation $\mathbf{h}$ does not follow a closed-form distribution. In terms of the training objective for our model, we have revised the formula and our prior and posterior distributions are on the two random variables $\mathbf{r}$ and $\mathbf{z}$.
>
> * Posterior over module selection variable
>
> This was a mistake on our side when writing the training objective. The posterior is indeed a conditional probability on the previous slot representations $\mathbf{h}$ and we are using the same network as described in section 3.2. We will update equation 5 to reflect this.
>
> * Independence of module selection prior
>
> The module selection prior  is conditional on previous module selections implicitly through being conditional on the previous slot representation  $\mathbf{h_{t-1}}$, which is itself a function of previous module selections $\mathbf{r}^k_{<t}$. $\mathbf{h_{t-1}}$indeed encodes all the past information about the slot (including how it has been modified by the previously selected modules and additive updates).
>
> * Mixture of gaussians observation model
>
> It is true that a mixture of logistics would probably provide a tighter ELBO by reducing the probability mass on continuous pixel values. Most previous object-centric approaches, including MONET, IODINE and Slot Attention, use mask decoders with a mixture of gaussians observation models for their simplicity. We round the continuous output of the mixture of gaussians, thus making sure that there are no invalid pixel values or values outside of the output range. Note that mixtures of logistics also have to be discretized. We are unsure of degenerate solutions that would be avoided by the mixture of logistics, and we would be grateful if you could provide additional details on this point.
>
> * Similarity to Switch Transformer
>
> The Switch Transformer is quite different from our model, its goal being to execute different subnetworks depending on the input example. In our case, the rule to apply to each slot is not a deterministic function of the input or context (i.e. the previous slot), but rather something that we either infer given a ground-truth observation of an observation $\mathbf{x}_t$ (posterior), or a random variable with a certain probability distribution given by the prior. It might be possible to select a rule with a key-query mechanism, but conceptually we believe it makes sense to compute all possible future states for a slot, and then select the right transition based on the similarity of the candidate state to the ground truth observed state.
>
> * Questions about the selection variable
>
> The module selection variable is a discrete variable. During training it is represented as a continuous relaxation of a one-hot vector, as a result of sampling from a Gumbel-Softmax distribution. Note that in this case Gumbel-Softmax does not use the straight-through estimator. At test time we sample from the softmax distribution and therefore obtain a discrete one-hot vector. We use a gumbel-softmax distribution as in general it provides better gradient estimates than using the straight-through estimator.

---

> > ### Author Response · Authors · 2021-12-03
> > **Reply (2)**
> >
> > * Inductive bias
> >
> > We would like to note that there is a difference between the observation distribution and the capacity of the model. Mixtures of logistics are only better at allocating probability mass for discrete outputs compared to mixtures of gaussians, usually resulting in slightly better ELBOs, but at their core they are still mixture models. The inductive bias in our model does not come from the observation model, but rather from the object-centric and dynamic rule-centric scene decomposition. While it is possible for a model with *high capacity* (regardless of the observation model) to render the full scene from a single slot, it is slot attention that usually prevents such solutions and in general correctly segments the scene into its different objects even if we have high capacity models.
> >
> > Regarding whether the transition from $\mathbf{h}_{t-1}$ to $\mathbf{h}_t$ is deterministic and therefore it can be captured by a single module, that only happens in settings where the future is a deterministic function of the past. The majority of video prediction datasets as well as real world scenarios are stochastic, and the main goal of our paper is to propose a stochastic counterpart to models such as SCOFF, which fail in stochastic settings (see our comment on comparision with  SCOFF and RIM that showcases this).
> >
> > * Updated Figure 3
> >
> > As stated in the general comment, we updated our main Figure 3 with improved mode-module matching colormaps (92% vs 80%). We realized that our learning rate decay was not optimal and resulted in suboptimal performance of VIM variations.
> >
> > * Minor comments
> >
> > Thank you for catching those. We will revise the citations and modify the figure accordingly.
> >
> > We hope this answer addresses your concerns and we would be happy to provide additional information if needed.

---

> > > ### Comment · Reviewer_hQZF · 2021-12-20
> > > **Response to rebuttal**
> > >
> > > I would like to thank the authors for annswering my questions. I have updated my score accordingly.
> > >
> > > Regarding the mixture of Gaussians: The problem of modeling the output with a Gaussian distribution is that if you parameterize the standard deviation, then the model can make it arbitrarly small for certain pixels. MONET gets around it by fixing the scale, while if you truly want to model a distribution over the image space, the mixture of discrete logistics discretizes the output space with full support while allowing to take derivatives.

---

### Official Review · Reviewer_4fRa · 2021-11-17

**Confidence:** 3
**Overall Score:** 5

**Main Review:**

Strengths:
The paper exploits ideas of sparsity and compositionality in latent space to recover mechanisms explaining the data that generalizes out-of-distribution.
The paper builds on previous work learning independent mechanisms and makes a variational version of them that appears meaningful. The initial experimental results reported are interesting and promising (but they could be more convincing, see below).

Weaknesses:
The ideas motivating the work and the principles behind the architecture are not particularly novel and follow closely previous related works: in particular RIM (Goyal 2019) and SCOFF (Goyal 2020).
The main difference with previous works is that the current paper proposes a generative model.
What problem does the generative VIM solve that was out-of-reach for RIM-kind of approaches?

This justification appears important because this paper focuses on toy 2D scenarios whereas SCOFF tackles more challenging evaluation setups. Also, this paper focuses on "unary" modules, whereas it seems that SCOFF can deal with more arbitrary relationships between modules and objects.

Thus, it seems that the paper should better justify the advantages brought by the variational mechanisms compared to these previous works. It would be interesting to also discuss (and report) what RIM and/or SCOFF would produce on the same 2D toy setup considered in this work.

The experiments are limited to a 2D setup with moving balls.
Furthermore, there is no real comparison to baselines and for each experiment, only one run is reported. There is no measure of performance across many test runs. Only Figures 4 and 5 have some notion of global performance, but there is no error bound and we don't know on how many runs this was computed.
In general, the reporting of experimental results could be more thorough and convincing. In particular, 5.2 has only a very shallow description of results.

Minor comments:
If only up/down/left/right are modeled, can the modules encode notions of velocity? Velocities would seem quite important to track objects moving in time but are more abstract than up/down/left/right.

In equation (2), shouldn't it be p(z_0|c) instead of p(z_0^k|c)?
The paragraph "module selection posterior" overflow in the margin
The experiments report results when varying the number of modules, what about varying the number of slots?

**Summary:**

This paper proposes Variational Independent Mechanism, a generative model trained to uncover a latent set of objects and disentangled mechanisms that act on these objects. It proposes a generative extension to previous works learning independent mechanisms and tests it on a toy 2D video setup.

---

> ### Author Response · Authors · 2021-12-03
> **Reply**
>
> Thank you for your valuable feedback. We answer your concerns below:
>
> * What problem does VIM solve compared to SCOFF and RIM?
>
> The main advantage of VIM over RIM and SCOFF is the ability to model stochastic sequences. RIM and SCOFF are deterministic models that assume that the future is a deterministic function of the past and trained only on a next step prediction task. Instead, VIM is a probabilistic model that can capture multimodal future outcomes. Most datasets and real world scenarios are multimodal, and therefore this is an important aspect for video prediction. Papers such as SV2P[1] and SVG[2] improved upon previous video prediction models by showing that only probabilistic models can model complex datasets such as BAIR Push. Similarly, VIM introduces a probabilistic model in the context of object-centric video prediction with independent mechanisms that can handle stochastic environments.
>
> * Dataset complexity
>
> While it is true that our random walk ball datasets are simple, we believe that they are adequate to show that VIM can model stochastic sequences and to serve as a proof of concept for initial research in this area. Compared to SCOFF, our datasets are of similar visual complexity. Only the RL environments in SCOFF have more complex 2D visuals, however those datasets are not used for video prediction tasks. Additionally, we have run a SCOFF baseline in our proposed dataset - SCOFF is unable to properly model sequences in this setting because it cannot model multimodal outcomes. Thus, despite the visual simplicity of our dataset, we believe it is complex enough to highlight the capabilities of VIM compared to SCOFF or RIM.
>
> * Report the performance of SCOFF and VIM on our proposed setup
>
> Thank you for proposing this baseline. Following your suggestion we have run a SCOFF and a RIM baselines in our proposed dataset. The results from this experiment can be found in a general comment for the paper. We observe that both models fail to produce sharp generations and instead model the mean next frame. This is due to SCOFF and RIM being deterministic models, and highlights the advantages of the variational formulation of VIM.
>
> * Multiple runs per experiment
>
> We have added 2 seeds per model for the main diagram results in Figure 3 (see main comment for an updated figure). We have also further improved our hyperparameter selection and re-ran the model with a better learning rate decay schedule. We have obtained better module interpretability and better reconstructions, and we will update the results accordingly in Figure 3 in the paper.
>
> * Minor comments
>
> Thanks for catching the typo in the math, we have updated Section 3 to fix this. Regarding the number of slots, if we do not use enough slots then we do not have good object separation (as in the case of slot attention). If using more slots than needed (which is our most common setup), then slot attention learns to assign blank slots that are not used.
>
> We hope our answer helps resolve your concerns and we would be happy to answer any further questions you might have.
>
>
> References:
>
> [1] ]SV2P: Babaeizadeh, Mohammad, et al. "Stochastic variational video prediction." 6th International Conference on Learning Representations, ICLR 2018. 2018.
>
> [2] SVG: Denton, Emily, and Rob Fergus. "Stochastic video generation with a learned prior." International Conference on Machine Learning. PMLR, 2018.

---

### Official Review · Reviewer_ShZb · 2021-11-22

**Confidence:** 3
**Overall Score:** 8

**Main Review:**

The reviewer have elementary familiarity with video prediction (graduate level textbook knowledge) and is fairly familiar with unsupervised representation learning.

Detailed Comments:
1)	Modules: I might have missed it, but what is the functional form of the modules (f_\theta)? In the experiments? And what are the common/possible choices in general?

2)	With regard to interpretability, I feel like many alternative ways can achieve similar level of interpretability (e.g. object tracking + unsupervised learning over the trace), so it might be useful to establish some benchmark or show that the module’s abstraction is better than some alternative.

3)	Dynamic Transfer: I feel like this is the most important results since this illustrate the benefit of disentangle the underlying dynamic rules of the generation process. I think it would be useful to compare VIM with prior methods such as SCOFF and GSWM.




**Summary:**

The authors introduced the VIM for learning latent representations of objects and their transition function overtime from video. I find the paper clearly written and the experiments well executed, however, I think if the results can be put in context of the performance of other method, it would be more illustrative

---

> ### Author Response · Authors · 2021-12-03
> **Reply**
>
> Thank you for your encouraging feedback! We answer your comments below:
>
> * Form of the modules
>
> The modules are implemented as 2-layer MLPs that take as input the previous slot representation $\mathbf{h}_{t-1}$ and produce a candidate current slot representation.
>
> * Benchmarks for Interpretability
>
> The main advantage of our approach is that i) it can model stochastic sequences and ii) does not require supervision. Approaches such as SCOFF or RIM do not require supervision but cannot model stochastic sequences since they are deterministic. On the other hand, most tracking approaches require some kind of supervision. We will think on how to establish a benchmark for stochastic and unsupervised object-centric sequence modelling that goes beyond the metrics of likelihood/ELBO.
>
> * Dynamic Transfer results
>
> For the settings explored, SCOFF cannot perform dynamic transfer because it is a deterministic model and fails to model the sequences in our setting. See our Comparison to SCOFF and RIM comment.
>
> We hope our answer helps resolve your comments and we would be happy to answer any further questions you might have.

---

### Author Response · Authors · 2021-12-03
**Comparison to SCOFF and RIM**

SCOFF proposes an architectural component to process a visual input such that it can be explained in terms $K$ object files (corresponding to slots in our model) and $M$ schemata (modules) processing those objects files.
They propose to test their architecture in several downstream tasks and one of them is video prediction.
To do so they train SCOFF on a next-step prediction task with an autoregressive formulation.
RIM corresponds to a SCOFF with $M = 1$ schema (in other words, with a single module). At each time step $t$ the computations are conditioned on the previous frame.

A key assumption in SCOFF and RIM is that the outcome is deterministic given previous frames.
Similar to SV2P and SVG-LP, we argue that this leads to blurry predictions in settings with stochastic dynamics, as the model is trying to capture the mean of all plausible future outcomes.
To overcome this limitation we propose to include the intuition behind SCOFF's architectural inductive biases in a variational framework, where the schemata would rather correspond to the different modes/rules of the stochastic dynamics of the environment.

We report in the table below the reconstruction MSE averaged over 5 time-steps sequences for SCOFF and RIM (with teacher forcing at each time step) and VIM (using its inference network) for our proposed random walk  balls dataset. All models are trained with 4 slots and VIM and SCOFF are trained with 5 modules/schemata.
We expect SCOFF and RIM to fail to produce sharp samples since they assume that the future is deterministic given the previous frame, whereas we expect VIM to handle the multimodality of the future outputs. This is reflected in the Reconstruction MSE, where VIM obtains significantly better results.

| Model         | Reconstruction MSE     |
|--------------|-----------|
| RIM           |  $91.7 \pm 5.4$ |
| SCOFF      | $81.9 \pm 1.0$ |
| VIM (Ours) | $\mathbf{4.2 \pm 0.5}$ |

Additionally, we add [this visualization of a SCOFF reconstruction](https://imgur.com/2mMoIVp). We notice that, while SCOFF correctly identifies the different objects in the scene, it cannot reconstruct frames and instead predicts the mean of all the plausible future frames.

---

### Author Response · Authors · 2021-12-03
**Summary of changes**

We would like to thank all reviewers for their useful feedback. As a result, we have run additional baselines and revised some parts of our submission. Below we provide a summary of the changes:

* Comparison to SCOFF and RIM

We have added RIM and SCOFF baselines to our main experiment. The results from this experiment can be found in the previous general comment, 'Comparison to SCOFF and RIM'. Both RIM and SCOFF are not able to model the stochastic sequences in our dataset, despite the visual simplicity. That is because SCOFF and RIM are deterministic models. We hope this experiment is useful to better understand the advantages of VIM and its performance compared to previous approaches in the literature.

* Updated math for Section 3

We have revised the model formulation. Before we had inconsistencies and the random variables in our model were not clearly described. To summarize, our model has two random variables: the module selection variable $\mathbf{r^k_t}$ and the additive update $\mathbf{z^k_t}$. The module selection variable $\mathbf{r^k_t}$ is discrete, and it is modeled with a Gumbel Softmax distribution. The additive update $\mathbf{z^k_t}$ is a continuous variable that follows a gaussian distribution. The slot representation is denoted by $h^k_t$, and it is computed as  $\mathbf{h_t^k}$ = LayerNorm( $\mathbf{h_{t-1}^k} + f_\theta(\mathbf{h}_{t-1}^k, \mathbf{r}_t^k) + \mathbf{z}_t^k) $.
$h^k_t$ is thus a deterministic function of random variables. We have prior and posterior distributions on both random variables $\mathbf{r^k_t}$ and $\mathbf{z^k_t}$, and our ELBO is a function of these variables. We detail the updated math in [this image](https://postimg.cc/ZBJTNHTn). We will include the revisions in Section 3 of the main text. We hope this clarifies our model formulation.

* Multiple runs and updated results

We have run multiple runs for our main experiment. Our model tends to converge to similar solutions with low variance, indicating that it is well optimized. There were only few runs where the model got stuck in a local optima where only a few modules are used and the reconstruction error is high. However, with careful hyperparameter tuning this effect can be avoided, specially when tuning the learning rate warmup and schedule, which was also important for slot attention to obtain good slot separation. Another important hyperparameter is the schedule for changing the length of the sequences that the model is trained on.
We will update the main results in Figure 2 and 3 where we used a better learning rate decay that resulted in overall better performance.
An updated Figure 3 can be found [in this link](https://imgur.com/AJsNHqi)

We hope these changes help resolve the concerns raised by the reviewers and we would be happy to provide additional clarifications if needed.

---

### Decision · Program_Chairs · 2022-01-12

**Decision:**

Accept (Poster)

**Comment:**

This paper is above the bar for acceptance. The authors rebuttal included new empirical studies that allayed the initial concerns.